# Measuring Happiness in Adolescent Samples: A Systematic Review

**DOI:** 10.3390/children9020227

**Published:** 2022-02-08

**Authors:** Justė Lukoševičiūtė, Gita Argustaitė-Zailskienė, Kastytis Šmigelskas

**Affiliations:** 1Department of Health Psychology, Faculty of Public Health, Medical Academy, Lithuanian University of Health Sciences, Tilžės g. 18, LT-47181 Kaunas, Lithuania; gita.argustaite-zailskiene@lsmu.lt (G.A.-Z.); kastytis.smigelskas@lsmu.lt (K.Š.); 2Faculty of Public Health, Health Research Institute, Medical Academy, Lithuanian University of Health Sciences, Tilžės g. 18, LT-47181 Kaunas, Lithuania

**Keywords:** happiness measurement, adolescent, mental health, systematic review

## Abstract

Background: Happiness is a phenomenon that relates to better mental and physical health and even longevity. There has been an increase in surveys assessing subjective well-being as well as happiness, one of the well-being components that reflect one’s feelings or moods. Happiness is mostly measured in adult samples. There is a lack of an overview of the tools used to evaluate adolescent happiness, so this paper aimed to review them. Methods: A literature search was performed in the PubMed and PsycArticles databases (2010–2019). In total, 133 papers met the eligibility criteria for this systematic review. Results: The results are grouped according to the type of measure, single or multiple items, that was used in a study. Almost half of the studies (64 of 133) evaluated subjective happiness using single-item measures. The most commonly used scales were the 4-item Subjective Happiness Scale and the Oxford Happiness Questionnaire. Among the 133 articles analyzed, 18 reported some validation procedures related to happiness. However, in the majority of cases (14 studies), happiness was not the central phenomenon of validation, which suggests a lack of happiness validation studies. Conclusions: Finally, recommendations for future research and for the choice of happiness assessment tools are presented.

## 1. Introduction

Being happy is the goal of many people’s lives. Interest in the phenomenon of happiness has been observed since ancient times. It is acknowledged that the philosopher Aristotle was the first to raise questions in the philosophical literature as to how happiness may best be understood [1]. Later, scientists in the field of philosophy defined happiness as “the belief that one is getting the important things one wants, as well as certain pleasant affects that normally go along with this belief” [2] (p.178). In 1999, Lyubomirsky and Lepper [3] proposed a different perspective on happiness, naming it “subjective happiness”, which is currently defined as the individual’s perception of being a happy or unhappy person [4].

During the last few decades, there has been a growing interest in positive psychology, especially concerning subjective well-being or happiness. Studies show that happiness can be influenced by both personal and external factors. The World Happiness Report [5] names income, work, community and governance, values, and religion as external factors, while mental and physical health, family experience, education, gender, or age may be regarded as personal factors.

Recent studies show that the relationship between happiness and health is developing rapidly, exploring the possibility that impaired happiness is not only a consequence of illness and health but also a potential risk factor [6]. The mechanisms potentially linking happiness with health include lifestyle factors, such as physical activity and nutrition [6]. A large-scale nationwide general population survey in China by Tan et al. [7] confirmed that happiness is associated with subjective health assessment and is more important with increasing age. Recent longitudinal research indicates that higher levels of happiness have numerous important health advantages, such as the reduced risk of mortality or morbidity in adults [6]. Moreover, increasing happiness could contribute to lowering health care expenses and, thus, have an impact not only on public health but also on the country’s economy [5,8].

The abovementioned factors are mostly evaluated and analyzed among adults. Children and adolescents, however, are much less investigated in this regard, even though their psychoemotional state is often crucial for their overall health in the absence of physical illness or disability. Here, longitudinal studies show that positive well-being (such as positive affect and self-esteem) during adolescence is associated with better perceived general health during young adulthood when controlling for depressive symptoms. Another large-scale cross-sectional study investigated adolescents and found that a low happiness level was strongly associated with sleep problems [9]. In addition, positive well-being is also significantly related to fewer risky health behaviors, such as insufficient physical activity, fast food, binge drinking, smoking, and the use of illegal substances [10,11]. Other studies also confirm that higher happiness scores protect adolescents against cigarette smoking [12]. Moreover, higher unhappiness can be observed in schoolchildren who have very poor or non-intact families [11]. Overall, happiness as a positive affect could be the key indicator of adolescent psychoemotional well-being, quality of life, or even future success [13]. These results suggest that a sufficient level of happiness at a young age could be a predictor of health behavior and later health outcomes. Therefore, the assessment of happiness and efforts to increase it should be an important public health concern, as it relates to both the current and future physical and mental health of a population.

Despite a large quantity of research on happiness, there still is a problem regarding its terminology. In both science and practice, the construct of happiness is often used interchangeably with life satisfaction, subjective well-being, quality of life, flourishing, or contentment [12,14,15].

Therefore, when considering happiness, one of the most problematic aspects is measurement. Many different tools are used to evaluate this construct, raising the question of a “gold standard” for the accurate measurement of adolescent happiness.

Recent examples of studies assessing adolescent happiness have used the Subjective Happiness Scale [16,17], the Oxford Happiness Questionnaire [18], the Oxford Happiness Inventory [19], or several variations of a single-item question about the respondents’ current degree of happiness [20]. However, some authors refer to certain tools as measures of happiness even though the instruments do not measure happiness as such (e.g., the WHO-5 Well-being Index ([21,22]). In some instances, authors use questions or subscales to measure happiness from larger-scope scales that are normally used to measure another subject, e.g., the Piers–Harris Children’s Self-Concept Scale [23] or the Profile of Mood States (POMS) questionnaire [24].

To our knowledge, there are some literature reviews about youth life satisfaction [25] and a systematic review on well-being measurements [26]. However, there is a lack of reviews about the tools used to evaluate adolescent happiness. In this context, the aim of this review is to fill that gap by reviewing the instruments that have been used to assess adolescent happiness over the last 10 years. In addition, we review happiness-related validation studies and the potential need for such validation.

## 2. Materials and Methods

### 2.1. Literature Search Strategy

The systematic review was carried out following the preferred reporting items for systematic reviews and meta-analyses (PRISMA) recommendations for systematic reviews [27]. A systematic literature search was conducted in February 2020 to identify all relevant studies published from 1 January 2010 to 31 December 2019. The search was performed in two electronic databases: PubMed and PsycArticles. To identify the studies on happiness, the search keyword “happiness” was combined (using the conjunction AND) with specifications of the age group: “adolescent”, “adolescents”, “adolescence”, “child”, “children”, “childhood”, “schoolchildren”, “school-children”, “youth”, “youngster”, “youngsters”, “teenager”, “teenagers”, “teen”, “teens”, “student”, “students”, “kid”, “kids”, “pupil”, “pupils”, and “juvenile”.

### 2.2. Selection Criteria

The eligibility criteria were developed by three reviewers (J.L., G.A.-Z. and K.Š.), who are all co-authors of the present article. The reviewing team consisted of experts in adolescent health and psychology. Reviewers independently applied the pre-defined inclusion and exclusion criteria for the retrieved articles.

#### 2.2.1. Inclusion Criteria

The inclusion criteria encompassed scientific papers in all languages that reported the assessment of adolescent happiness and were published in peer-reviewed journals. Only the articles with the full text available were included.

#### 2.2.2. Exclusion Criteria

The exclusion criteria were as follows: duplicate search results, studies having no adolescent participants (our target group was 11 to 17 years old), non-empirical studies, review papers (e.g., systematic or another reviews and meta-analyses), qualitative studies, papers not specifying the method for measuring happiness, studies addressing a specific aspect of happiness (such as school-, body-, family-, pregnancy-, or oral-health-related happiness) as opposed to general happiness, and studies where happiness was not self-reported by adolescents (e.g., assessed by their parents or teachers).

### 2.3. Study Selection Process and Data Extraction

First, a literature search was conducted in the scientific databases by combining keywords. Afterwards, we identified and excluded the duplicated articles. After this first selection, the titles and abstracts of the articles found were reviewed. Next, a second exclusion process was made of those studies that did not fulfill the inclusion criteria. The articles obtained after this last selection were evaluated in depth to check for the specific inclusion criteria. Finally, the studies that form part of this review were identified. Each article was reviewed independently by two of the three investigators (J.L., G.A.-Z. and K.Š.) with an almost equal overall number of reviewed papers per investigator. Discrepancies between the reviewers were resolved by discussion and consensus. Using a structured template by all reviewers, the following information was retrieved and summarized: publication’s author(s), year, country of study, design of the study, sample (total size and gender ratio), sample type (clinical or non-clinical), response rate, and happiness measurement tool (scale, subscale or question, response options, and internal consistency).

The data were extracted from the original articles. In the case of insufficient data or unclear reporting, the original articles’ authors were contacted by email. In total, 18 authors were contacted and 11 of them provided additional information for use in this review.

### 2.4. Assessment of the the Methodological Quality of the Studies

These criteria were developed due to the lack of existing quality assessment checklists for different study designs. According to our knowledge, many checklists were developed for the quality assessment of randomized controlled trials, cohort studies, or case-control studies. However, the tools for observational epidemiological studies in general [28] and cross-sectional studies in particular are not well accepted [29], even though they comprise the largest part of the studies under this systematic review. Our criteria were selected based on previous review studies about methodological quality assessment [30] and already developed tools, such as The Newcastle-Ottawa Scale (NOS) for assessing the quality of non-randomized studies in meta-analyses [31], the NIH quality assessment tool [32], and the CASP (Critical Appraisal Skills Programme) [33] checklists.

The quality of the studies included in this systematic review were assessed using 5 criteria with 1 to 3 quality stars, giving scores ranging from a minimum of 5 stars to a maximum 15 stars (Table 1). The quality score was used to indicate the strength of the evidence from the individual studies but was not used to determine their inclusion or exclusion within the review. The methodological quality assessment was performed independently by all three authors (J.L., G.A.-Z. and K.Š.), with each article being reviewed by two researchers. The third researcher was consulted in the case of discrepancies. Since this study did not investigate the effectiveness of interventions, neither a risk of bias evaluation nor a meta-analysis were performed.

## 3. Results

### 3.1. Study Selection

In total, the initial search yielded 12,808 records of scientific publications. Once duplicates were removed, 2057 publications were identified. These publications were screened according to the eligibility criteria in the titles, abstracts, and full texts. Figure 1 shows a PRISMA flow chart depicting the articles’ identification, screening, eligibility, and the inclusion process.

The titles and abstracts of the publications were analyzed using the inclusion and exclusion criteria, and 1057 articles were excluded. Furthermore, 1000 articles were reviewed and 867 of them were excluded after accessing the full texts, leaving 133 papers in the review. The majority of the articles were excluded due to the age of the sample (age group outside of the 11–17 years range).

### 3.2. Characteristics of the Included Studies

An analysis of the studies by country revealed that adolescent happiness was most frequently investigated in the UK (17 studies of 133), 12 studies were conducted in South Korea, and 11 were conducted in the USA. The majority of the studies were conducted in Europe and Asia (mostly east Asia). Moreover, there were 12 international studies found that covered up to 109 countries. The main characteristics of the included studies are shown in Appendix A (Table A1).

More than half of the articles (*n* = 88) had a sample size above 500 (median *n* = 1165). Usually, the group of interest was a general population of adolescents (*n* = 120), drawn based on randomization, convenience, or school sampling. Some studies investigated adolescents with clinical conditions (*n* = 10) or compared them with non-clinical subgroups (*n* = 3). The majority of the studies used a cross-sectional design (*n* = 101). More rarely, the studies were longitudinal (*n* = 23) and several studies had experimental or twin designs. Mostly (*n* = 108), the gender distribution was approximately balanced. According to the participants’ age, more than one-third of the studies (*n* = 57) analyzed only adolescents aged 11–17 years. However, some studies included not only adolescents but children and adults as well, and their findings were not always separated by age group.

### 3.3. Study Quality Assessment

All included studies had a generally moderate to high quality, ranging from 7 to 15, with median of 11 and a visually normal distribution (Table 2). The highest medians (3 points) were observed for the criteria of gender balance and sample size. Lower medians (2 points) were observed for representativeness and measurement, while the lowest median was observed for the response rate (1 point). The study quality assessments for each criterion of all included studies is shown in Appendix B (Table A2).

### 3.4. Happiness Assessment Tools

#### 3.4.1. Single-Item Happiness Measures

Almost half of the studies under review (64 of 133) evaluated subjective happiness using single-item measures. The questions addressing the perception of happiness were mainly concerned with how the participants perceived the present moment or current period (*n* = 20). Some asked about life in general (*n* = 15) and many studies did not specify the approach (*n* = 25) (Figure 2). Several studies employed a specific time to describe the perception of happiness, such as the last week [125], last two weeks [157], or last month [146] (Appendix C, Table A3).

It should also be noted that the majority of the studies with single-item measures (*n* = 41) formulated the item more towards one’s feelings or perceptions of the self as happy, e.g., “In general, are you happy lately?” [101] or “How would you say you feel these days?” [96], while others (*n* = 16) focused on the perception of a happy life rather than self, such as “I feel happy about life” [76] or “In general, how do you feel about your life at present?” [156].

Studies with single-item measures used various ranges of responses options to the question. More frequently the studies had an odd number of responses than an even number (37 versus 24). When comparing the range of response options, there was a clear preference for 4 (*n* = 18) or 5 response options (*n* = 16) rather than any other range.

In total, 56 of 64 articles on studies with single-item measures of happiness explicitly described the response options, which were mostly framed as numerical scales with labeled captions to go along with the numbers. Overall, 35 studies defined labels with every ordinal response option (e.g., “very happy”, “happy”, “average”, “unhappy”, and “very unhappy”), with only two of the studies using a dichotomous measure [74,76]. Several studies (*n* = 7) used continuous scales with only the lowest and highest values labeled, and this was usually seen with response options in the 10- or 11-point range. There was an overall trend that labeled responses were preferred for shorter ranges and longer ones were not labeled.

#### 3.4.2. Multiple-Item Happiness Scales

The summarized results of the happiness questionnaires used for adolescents are presented in Table 3. The most commonly used scale was the 4-item Subjective Happiness Scale (SHS) [3], which was used in 24 studies. All other scales were used much more rarely, the most common of them being the Oxford Happiness Questionnaire (OHQ) [161] with longer 29-item (*n* = 7) and shorter 8-item (*n* = 5) versions. Overall, it was observed that validated questionnaires were chosen more frequently than unique scales. Newer questionnaires tended to be shorter, including less than 10 items. Detailed information on the studies, by scale and their internal consistency, is presented in Appendix D (Table A4).

#### 3.4.3. Happiness as a Part of Other Scales

Some studies measured happiness using some subscales from the tools that measure concepts larger than happiness (Table 4). The most commonly used scales of this kind were the Piers–Harris Children’s Self-Concept Scale (*n* = 5) and the Profile of Mood States (POMS) (*n* = 4). Most frequently, the happiness items (subscales) comprised the minor part in such scales. There were some inconsistencies with subscale use, where some studies chose to use particular sets of items for happiness, even though such items were measuring not only happiness but also feeling joyful, calm, cheerful, or satisfied. Detailed information on the studies, by subscale and their internal consistency, is presented in Appendix E (Table A5).

### 3.5. Validation of Happiness Measures

Among the analyzed 133 articles, 18 reported some validation processes related to happiness. However, in the majority of cases (*n* = 14), happiness was not the central phenomenon of interest, it was a subject for the validity of other constructs. It should be noted that virtually all validation procedures included either convergent or concurrent validity and very rarely included any other type of validity (Table 5). None of the studies addressed discriminant or predictive validity.

The specifically validated happiness tools were the Oxford Happiness Inventory [46], the Oxford Happiness Questionnaire [159], the Pemberton Happiness Index [22], and a single-item measure [90]. In these cases, mainly convergent validation was assessed, associating happiness with other psychological well-being phenomena, such as life satisfaction, love of life, or positive affect conditions (e. g. calm and peaceful). Such correlations were mainly medium-sized. Studies with comparisons of different happiness measures were very scarce. One study showed that the Oxford Happiness Inventory correlated with a single-item happiness measure (range 0 to 10) at r = 0.57 [46]. Another study revealed that the Pemberton Happiness Index has incremental validity compared to the Subjective Happiness Scale (r varied from 0.69 to 0.83), though none of the items of the former directly addressed happiness as such [22].

All other studies that conducted validity calculations related to happiness were mainly focused on the validity of other phenomena of interest, such as well-being [53], emotional competences [37,49], mental health [59,141], personality [124], and other concepts. In contrast to the happiness-centered validation studies, here, concurrent validities were also calculated, apart from convergent. Happiness was related to other measures at small or medium correlation levels. For instance, it was related to lower neuroticism [113,124], higher mindfulness [139], less depression [135,136], and a better coping response [64]. It can also be noted that one study evaluated incremental validity [22].

Altogether it can be seen that the validation studies in the recent research on adolescent happiness are quite limited, mainly addressing other constructs associated with positive affect. There is a lack of studies with predictive validity and test-retest reliability, as well as comparisons of the different happiness tools.

## 4. Discussion

Happiness, as a phenomenon, has been a much-debated topic, investigated from the perspectives of different scientific disciplines. The current review focused on the psychological perspective. Presently, subjective well-being is once again a topic of interest to many researchers in psychology, and as a consequence, there is a growing body of research on happiness as its component.

Subjective well-being is a complex construct that refers to optimal psychological functioning and experience. It can be categorized into hedonic (the attainment of pleasure and avoidance of suffering), eudaimonic (meaning and purpose in life) [173], and evaluative (evaluations of how satisfied people are with their lives) [174] aspects. Happiness is considered as a part of the hedonic aspect, sometimes referred to as affective well-being [6].

This theoretical perspective makes it clearer why some researchers refer to happiness when, in fact, they measure life satisfaction, which is a construct that represents evaluative rather than affective well-being. The current systematic review also revealed this ambiguity in terminology of happiness. We found several studies in which the authors claimed they had measured and reported happiness while, in fact, they had assessed another, usually larger, aspect of subjective well-being [22,46].

The current analysis included a detailed review of the instruments that assessed happiness to see how this construct was operationalized across studies. While measuring happiness among adolescents, the hedonic perspective was predominant. Items in scales usually approached the participants’ current sense or perception of happiness. There was some variation as to the momentary or general sense of happiness, the latter including a time definition (such as two weeks) or simply the words “usually”, “overall” or “in general”. Most single-item measurements focused on a person’s feelings and asked a fairly direct question about their sense of happiness, e.g., “In general, are you happy lately?” [101]. This method gives researchers a clear answer about one’s happiness through the face validity.

Meanwhile, when analyzing multiple-item happiness scales, not many direct questions about happiness were observed. For example, the Oxford Happiness Questionnaire [161] contains one direct statement (out of 29 items) related to happiness: “I am very happy”. Another example is the Happiness Measures [165], which assesses feelings of happiness related to various life domains: the house or apartment, the people one lives with, the people in one’s family, friends, etc. (16 aspects in total). In this way, many various aspects of life are evaluated by asking whether the child feels happy about them. Shorter happiness scales, such as the Subjective Happiness Scale [3], are more oriented to the person and their personal feelings. It gives a shorter and clearer answer but does not assess separate aspects of happiness. Shorter scales result in a lower cognitive load for study participants, which is especially convenient when the target group is young. However, when happiness is a central phenomenon in a study, a validated happiness scale (possibly with subscales) seems to be a more suitable choice, given that psychometric characteristics are appropriate for the selected age group.

The overview of the tools also showed a clear emphasis on happiness as an affective aspect of well-being. Consequently, validity studies addressing happiness were also mainly targeted towards associations with other aspects of subjective well-being, especially with affective states and perceptions. Such affective constructs included depressiveness, neuroticism, emotional expressions, self-esteem, mindfulness, etc. Some validation studies also correlated happiness to cognitive or evaluative aspects of well-being, such as life satisfaction or perceived health [22], and, very rarely, to eudaimonic measures [53]. However, the eudaimonic aspect of well-being is more complicated to measure, especially among younger people, because it requires more extensive cognitive processing [6]. After all, the core of eudaimonic well-being, the meaning or purpose of life, is a presumably less relevant topic in children and adolescents than in adults.

In general, this review showed that in the last decade there was no abundance of validation studies on happiness. The ones we included in our review were mainly designated to validate other constructs, with happiness chosen as a non-central indicator. Specifically, the validated measures were the Oxford Happiness Inventory [46], the Oxford Happiness Questionnaire [159], and one single-item measure [90]. The majority of the scales being used in recent research were developed several decades ago. Consequently, it could be worth revisiting the validation in current adolescent samples. It should also be noted that none of the studies measured test-retest reliability to check how consistent the responses on happiness were over time.

The majority of studies using a single item for happiness suggest a continuum with four or five response options. This raises the question of the middle-point: an even number of choices (no middle-point) might force a person to report feeling either happy or unhappy, while an odd number with a middle point allows a responder to choose a doubting or indifferent state or opinion, which does not present the pressure to be happy, even though there may be some cultural preferences [175]. When including an even or odd number of responses, it is also worth considering the similarity with other items in the survey. It is likely better to follow the consistency of other items to decrease cognitive load.

### 4.1. Strengths and Limitations

To the authors’ knowledge, this paper represents the most comprehensive attempt to review the measurements that have been used to assess adolescent happiness. This systematic review includes the most recent articles published over the last decade (2010–2019). Moreover, it highlights the trends on which tools are used to assess adolescent happiness, as well as categorizes the measurements by their structure, frequency of use, and scale characteristics. This analysis revealed a lack of validation studies on happiness, which shows the need for such studies, given that many scales are relatively old and, therefore, may be questionable for contemporary adolescents.

However, this review also has some limitations. First, the literature search was performed in two databases. Only articles with an English summary were included in the search, which may have limited the results. However, a lot of duplicates were noticed and removed in the process, which lets the authors assume that not much additional information was lost. Nonetheless, the selected search databases, PubMed and PsycArticles, cover the fields of psychology and biomedicine and are widely used. We suggest that the articles published in journals outside of scope of those databases follow a general trend of scales’ use or otherwise choose local variations of happiness measurements that have low potential for international applicability. Moreover, this analysis omitted studies that evaluated an induced sense of happiness, e.g., happiness provoked during an experiment with certain stimuli. This exclusion criterion was chosen because our object was not an aroused, induced, or somehow provoked perception but rather a stable and overall sense of happiness as a state. There were also some studies, mainly experimental studies on emotions, that addressed happiness not as a specific phenomenon but rather as one of the emotional states. For example, children were asked to assess the drawings of facial expressions and then indicate if they felt sad, angry, happy, or scared [176]. Because such studies do not approach happiness per se but happiness as an aroused state within an emotional continuum, such studies were not included in this systematic review. Finally, among the studies included in this systematic review, there were also several covering broader age groups that included adults or children in addition to adolescents. It follows that the results of some studies may not be fully specific to our target group (adolescents).

Given that our review has covered the period from 2010 to 2019, we conducted an extra short overview of the articles from the most recent studies of 2020 and 2021. The vast majority of the studies used the previously administered questionnaires and scales of happiness. Several studies used validated language versions of EPOCH (Swedish [177]), or SHS (Brazilian [178]. In addition, one new scale, HERO, was developed [179] that assesses not only happiness but also enthusiasm, resilience, and optimism and is, thus, a general positive affect or well-being scale rather than a happiness measure per se. Other studies either used previously established scales or approached happiness as a validation tool for another constructs.

### 4.2. Suggestions for the Choice of Happiness Measurements

We suggest using specific tools (items or scales) for happiness instead of broader approaches. Moreover, when selecting the scale for measuring happiness, it is important to choose the one that specifically measures happiness and not just related constructs of well-being.The most common tools to measure happiness in adolescents are single-item measures, the Subjective Happiness Scale, and the Oxford Happiness Questionnaire. For the choice, it is important to decide on the length of the questionnaire. In studies where happiness is one of the constructs among others, a better option would be to choose either a single item or a short scale (e.g., the 4-item SHS) because other happiness scales are relatively long, which may be an obstacle for younger samples.If a single-item measure is chosen, the critical task is to decide on its range (four or five response options are used most commonly) and whether to use a continuous (visual-analogue) scale or labeled response options. In addition to that, when choosing a single item, it is relevant to keep in mind odd or even number responses. In the case of an odd number, there is an opportunity for the responder to choose the middle-point option without feeling forced to report being happy or unhappy.To increase the comparability of different studies’ findings, it is relevant to have approximately similar measurement tools addressing more or less the same construct. In the case of single-item happiness measures, it can be seen that some studies ask about the general, overall sense of happiness, while others specify the timing or situations.

## 5. Conclusions

In conclusion, this review provides a summary of the commonly used measures for assessing adolescent happiness. The research on happiness uses a variety of methods and instruments. About half of the studies included in the review assessed happiness using a single item, mostly employing statements that explicitly refer to being happy or the sense of happiness. This seems to be the gold standard in happiness research with single-item measures. Some studies refer to happiness even when measuring broader concepts related to subjective well-being. Other studies use more detailed approaches, providing data based on scales with multiple items. In contrast to the single items, however, such scales are much more diverse in covering not only the exact sense of happiness but also assessing different aspects of well-being and positive affect.

## Figures and Tables

**Figure 1 children-09-00227-f001:**
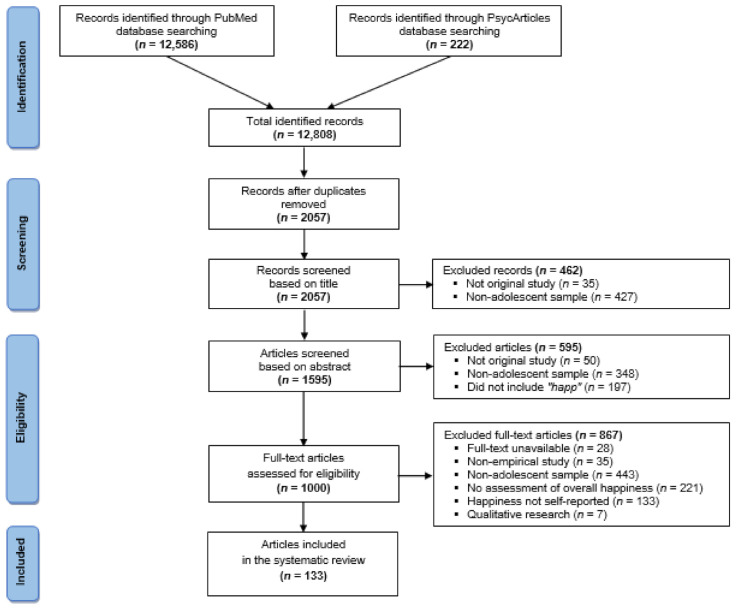
PRISMA flow chart of the systematic literature review.

**Figure 2 children-09-00227-f002:**
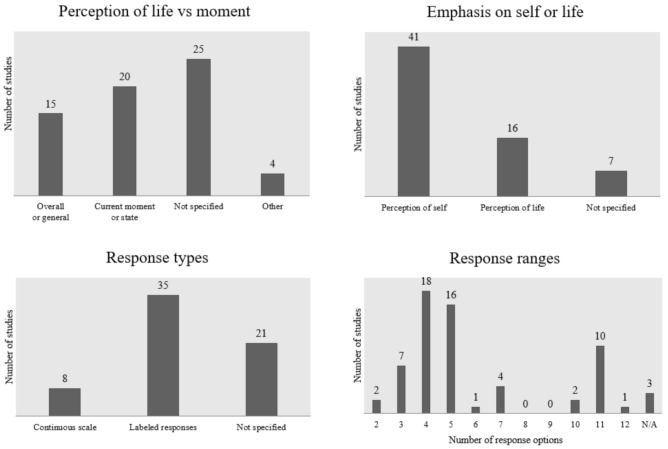
The main features of the single-item measures.

**Table 1 children-09-00227-t001:** Description of criteria for the methodological quality of the articles under review.

Criterion	3 Stars	2 Stars	1 Star
Representativeness of sample	Yes, representative	Not defined as representative, but coming from the general population	Selective, convenient, and similar sample or no data
Response rate	At least 80%	50% to 80%	Below 50% or no data
Gender balance	Difference between genders of less than 20%	Difference between genders of 20–50%	Difference between genders of more than 50% or no data
Sample size	At least 1000	100 to 1000	Less than 100
Measurement tool	Internationally used scale or subscale of happiness	Single item, including those drawn from other scales	Unique set of items and undefined scale

**Table 2 children-09-00227-t002:** Description of the criteria for the methodological quality of the articles under review.

Total Quality Score	Number of Studies	Reference
7	2	[34,35]
8	10	[36,37,38,39,40,41,42,43,44,45]
9	23	[23,46,47,48,49,50,51,52,53,54,55,56,57,58,59,60,61,62,63,64,65,66,67]
10	23	[24,68,69,70,71,72,73,74,75,76,77,78,79,80,81,82,83,84,85,86,87,88,89]
11	20	[4,90,91,92,93,94,95,96,97,98,99,100,101,102,103,104,105,106,107,108]
12	31	[22,109,110,111,112,113,114,115,116,117,118,119,120,121,122,123,124,125,126,127,128,129,130,131,132,133,134,135,136,137,138]
13	12	[11,139,140,141,142,143,144,145,146,147,148,149]
14	9	[12,150,151,152,153,154,155,156,157]
15	3	[158,159,160]

**Table 3 children-09-00227-t003:** Happiness scales and their structural information.

Scale	Scale Author(s), Year	Number of Items	Dimension(s)	Number of Studies
Subjective Happiness Scale (SHS)	Lyubomirsky and Lepper 1999 [3]	4	Unidimensional	24
Oxford Happiness Questionnaire (OHQ)	Hills and Argyle, 2002 [161]	29	Unidimensional	7
Oxford Happiness Questionnaire–Short Form (OHQ-sf)	Hills and Argyle, 2002 [161]	8	Unidimensional	5
Oxford Happiness Inventory (OHI)	Argyle et al., 1989 [162]	29	(1) satisfaction with life, (2) mastery and self-fulfillment, (3) social cheerfulness, (4) vigor, and (5) social interest	5
[scale title undefined in the article]	Chan and Koo, 2011 [163]	6	Unidimensional	3
Pemberton Happiness Index	Hervás and Vázquez, 2013 [22]	11	(1) remembered well-being (general, hedonic, eudaimonic, and social well-being), (2) experienced well-being (i.e., positive and negative emotional events that possibly happened the day before)	1
Humboldt Happiness Scale–Adolescent Version (HHSAV)	Reynolds, 2005 [164]	28	Unidimensional	1
Happiness Measures (HM)	Fordyce, 1988 [165]	2	Unidimensional (but the study in this review was used as multidimensional)	1
Gross National Happiness Abridged Survey (GNHAS) questionnaire	Pennock and Ura, 2012 [166]	48	psychological well-being, health, education, culture, time use, governance, community vitality, ecological diversity resilience, and living standards	1
WHO-5 Well-being Index	World Health Organization, 1998 [21]	5	Unidimensional	1
[scale title undefined in article]	Quy, 2019 [64]	9	Unidimensional	1
[scale title undefined in article]	Schacter and Margolin, 2019 [65]	3	Unidimensional	1

**Table 4 children-09-00227-t004:** Happiness subscales from the validated scales.

Scale	Version	Author(s), Year	Subscales	Number of Happiness Items of Total Items	Number of Studies
Piers–Harris Children’s Self-Concept Scale (PHC-SCS)	Piers–Harris Children’s Self-Concept Scale	Piers and Harris, 1963 [167]	Behavior, Intellectual and School Status, Physical Appearance and Attributes, Anxiety, Popularity, and Happiness and Satisfaction	10 of 80	4
	Piers–Harris 2 Children’s Self-Concept Scale	Piers and Herzberg, 2002 [168]	Behavioral Adjustment, Intellectual and School Status, Physical Appearance and Attributes, Freedom from Anxiety, Popularity, and Happiness and Satisfaction	10 of 60	1
Profile of Mood States (POMS)	Profile of Mood States questionnaire	McNair et al., 1971 [169]	Anger, Confusion, Depression, Fatigue, Tension, and Vigor. The modifications (by Kiang and Buchanan in 2013 and Mercado et al. in 2019 include Happiness (joyful, happy, and calm).	3 of 65	3 *
	Adolescent version (POMS-A)	Terry et al., 1999 [170]	Anger, Confusion, Depression, Fatigue, Tension, and Vigor	2 of 24	1 **
EPOCH measure of Adolescent Well-Being		Kern et al., 2016 [57]	EPOCH Measure of Adolescent Well-Being, which assesses five positive psychological characteristics (Engagement, Perseverance, Optimism, Connectedness, and Happiness)	4 of 20	4
Daily Mood Scale, an Internet version of the Electronic Mood Device		Hoeksma et al., 2000 [171]	Happiness (glad, happy, and cheerful), anger (angry, cross, and short-tempered), anxiety (afraid, anxious, and worried), and sadness (sad, down, and dreary)	3 of 12	3
Positive and Negative Affect Scale	For Children (PANAS-C)	Laurent et al., 1999 [172]	1 item on ‘happy’ as a part of the Positive Affect subscale	1 of 30	2 ***

* One study used a selected set of 19 items [24], another used 9 items comprising 3 subscales [99], and yet another used 10 items within 2 subscales [104]. ** The study used a selected set of 8 items from the original 24-item scale. *** modified versions.

**Table 5 children-09-00227-t005:** Happiness-related validation studies.

Study	Center	Happiness Scale	Construct Validity: Convergent	Criterion Validity: Concurrent	Content Validity
Abdel-Khalek, 2011 [46]	Happiness	Oxford Happiness Inventory (OHI)	Happiness correlates with other positive affect measures (love of life scale, life satisfaction scale, the mental health item, and the life satisfaction item).		Correlation between two happiness measures
Ali et al., 2012 [90]	Happiness	Single item	Happiness correlates with being calm and peaceful (rho = 0.43), lots of energy (rho = 0.37), full of life (rho = 0.48), the 4-item composite score (rho = 0.70), and with IQ.		
Lung and Shu, 2020 [159]	Happiness	Oxford Happiness Questionnaire (OHQ)	Happiness associates with psychological well-being and social adaptation.		
Hervás and Vázquez, 2013 [22]	Well-being	Pemberton Happiness Index	Happiness associates with different aspects of well-being and life satisfaction.	Happiness associates with sleep quality and perceived health.	
Brasseur et al., 2013 [49]	Emotional competence	Subjective Happiness Scale (SHS)		Happiness correlates with overall emotional competence (r = 0.40).	
Chen et al., 2012 [37]	Emotional expression and Gratitude	Single item	Happiness correlates with gratitude (r from 0.31 to 0.46) and ambivalence over emotional expression (r from −0.13 to −0.18).		
Cooper et al., 2011 [113]	Neurotic symptoms	Single item	Happiness associates with neuroticism.		
de Bruin et al., 2011 [139]	Mindfulness	Subjective Happiness Scale (SHS)	Happiness correlates with mindful attention awareness (r = 0.33).		
Disabato et al., 2015 [53]	Hedonic and eudaimonic well-being	Subjective Happiness Scale (SHS)	Happiness correlates with hedonic and eudaimonic well-being.		
Lardon et al., 2016 [100]	Wellness	Single item	Happiness associates with different well-being measures.		
Mahfoud et al., 2011 [59]	Mental health	Single item	Happiness associates with better mental health.		
Meleddu et al., 2012 [124]	Personality inventory and Self-esteem	Oxford Happiness Inventory (OHI)		Happiness correlates with extraversion (r = 0.48), neuroticism (r = −0.53), and self-esteem (r = 0.63) but not psychoticism (r = −0.04).	
Fat et al., 2016 [141]	Mental well-being	Single item		Happiness correlates with well-being (rho = 0.53).	
Quy et al., 2019 [64]	Coping response	9 selected items		Happiness associates with coping (d of separate items 0.2 to 1.0).	
Saarikallio et al., 2016 [43]	Music perception	Single item		Happiness correlates with the perception of healthy music (r = 0.21) and unhealthy music (r = −0.38).	
Salavera et al., 2017 [84]	Mind-wandering	Subjective Happiness Scale (SHS)	Happiness correlates with mind-wandering (r = −0.30).		
Yu et al., 2011 [135]	Depression	Single item	Happiness correlates with depressiveness (rho = −0.32).		
Yu et al., 2012 [136]	Depression	Single item	Happiness correlates with depressiveness (r = −0.41).		

## Data Availability

Not applicable.

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
