# Peer review of "Measuring Happiness in Adolescent Samples: A Systematic Review"

_children, 2022, doi:10.3390/children9020227_

Round 1
Reviewer 1 Report
This paper reports on a systematic review of istruments used to rate happiness among adolescents. The link between happiness and health is well documented, and the rationale for the study is strong. Assessing happiness during development shoud be an important aspect of child research, even though the number of studies addressing this topic are relatively few. This paper is therefore a useful contribution to the literature.
The manuscript is well organized and easy to follow in its various parts. The methods of the review are consistent with the guidelines for systematic reviews. The tables are informative and offer many details without being too dense.
I have only relatively minor comments:
- The decision to include publications from 2010 and 2019 should be better justified. Are previous reviews available to cover the topic prior to 2010? Are there other publications after 2019 that might be useful and should be included?
- I was surprised that there was an 84% duplication in step 2 of the review so that one goes from 12,808 down to 2057 publications (Fig.1). That seems quite high, though not implausible. Is it correct?
- On Tab. A1 (p. 6), Canada is misspelled as Kanada.
Author Response
|
The decision to include publications from 2010 and 2019 should be better justified. Are previous reviews available to cover the topic prior to 2010?
Are there other publications after 2019 that might be useful and should be included? |
Thank you for this point. Actually, the search process of this systematic review was conducted in early 2020, therefore the period of prior 10 years was arbitrary chosen. We have not found previously conducted systematic reviews on this topic but still wanted to review relatively recent studies in the field. For keeping up to date, we included an additional paragraph on the most recent studies on adolescent happiness from 2020–2021 that were not a part of our review: “Given that our review has covered the period from 2010 to 2019, we conducted an extra short overview of the articles from the most recent studies of 2020 and 2021. The vast majority of studies use the previously administered questionnaires and scales of happiness. Several studies have validated language versions of OHS (Chinese [177]), EPOCH (Swedish [178]) or SHS (Brazilian [179]. Also, one new scale was developed – HERO [180], assessing not only happiness, but also enthusiasm, resilience, and optimism, being thus a general positive affect or well-being scale rather than a happiness measure per se. Other studies either used previously established scales or approached happiness as a validation tool for another constructs.” |
|
I was surprised that there was an 84% duplication in step 2 of the review so that one goes from 12,808 down to 2057 publications (Fig.1). That seems quite high, though not implausible. Is it correct? |
Yes, it is correct because an average article was found using several synonymous combinations of keywords (as it may be seen, about 6 distinct combinations per article). |
|
On Tab. A1 (p. 6), Canada is misspelled as Kanada. |
Thank you – and apologies for our misspelling. We also amended several other inconsistencies in the countries’ list. |
Reviewer 2 Report
Thank you for letting me review this paper. The PRISMA methodology I'm familiar with, and it is well implemented and quite well described. I have no major issue with this paper being published, except for one thing: the two databases that were analysed are of a medical type only. Obviously, the authors are from this field, and the reasoning for using these databases is explained in section 4. However, other studies and analyses from other fields, like education, or generalist databases like Scopus /WOS, might provide complementary perspectives.
Author Response
|
The two databases that were analysed are of a medical type only. Obviously, the authors are from this field, and the reasoning for using these databases is explained in section 4. However, other studies and analyses from other fields, like education, or generalist databases like Scopus /WOS, might provide complementary perspectives. |
We appreciate your comment. Acknowledging this, we added an additional sentence on this issue in Discussion’s part on limitations: “We suggest that the articles published in journals outside of scope of those databases follow a general trend of scales’ use or otherwise choose local variations of happiness measurements that have low potential for international applicability.” |